# Prognostic Factors for Staying at Work for Partially Sick-Listed Workers with Subjective Health Complaints: A Prospective Cohort Study

**DOI:** 10.3390/ijerph17197184

**Published:** 2020-09-30

**Authors:** Kristel Weerdesteijn, Frederieke Schaafsma, Karin Bonefaas-Groenewoud, Martijn Heymans, Allard Van der Beek, Johannes Anema

**Affiliations:** 1Department of Public and Occupational Health, Amsterdam Public Health Research Institute, Amsterdam UMC, Vrije Universiteit Amsterdam, Van der Boechorststraat 7, 1081 BT Amsterdam, The Netherlands; f.schaafsma@amsterdamumc.nl (F.S.); karin.bonefaas@uwv.nl (K.B.-G.); a.vanderbeek@amsterdamumc.nl (A.V.d.B.); h.anema@amsterdamumc.nl (J.A.); 2Research Center for Insurance Medicine (KCVG), PO Box 7057, 1007 MB Amsterdam, The Netherlands; 3Department of Social Medical Affairs (SMZ), the Dutch Social Security Institute: The Institute for Employee Benefits Scheme (UWV), La Guardiaweg 94-114, 1043 DL Amsterdam, The Netherlands; 4Department of Epidemiology and Data Science, Amsterdam Public Health Research Institute, Amsterdam UMC, Vrije Universiteit Amsterdam, Van der Boechorststraat 7, 1081 BT Amsterdam, The Netherlands; mw.heymans@amsterdamumc.nl

**Keywords:** longitudinal, medically unexplained physical symptoms, paid work, remaining employed, sickness absence, work maintenance

## Abstract

Examination of prognostic factors for staying at work for long-term sick-listed workers with subjective health complaints (SHC) who partially work in a paid job, and to evaluate whether these factors are comparable with those of workers with other disorders. We used data of 86 partially sick-listed workers with SHC (57 females, 29 males, mean age 47.1 years) and 433 with other disorders (227 females, 206 males, mean age 50.9 years), from an existing prospective cohort study consisting of 2593 workers aged 18–65 years and registered as sick-listed with different health complaints or disorders for at least 84 weeks in the database of the Dutch Social Security Institute. We performed univariable logistic regression analyses (*p* ≤ 0.157) for all independent variables with the dependent variable staying at work for the workers with SHC. We then performed multivariable logistic regression analyses with forward selection (*p* ≤ 0.157) and combined the remaining factors in a final, multivariable model (*p* ≤ 0.05), which we also used for logistic regression analysis in the workers with other disorders. The following factors were significant prognostic factors for staying at work for workers with SHC: full work disability benefits (odds ratio (OR) 0.07, 95% confidence interval (95% CI) 0.01–0.64), good mental health (OR 1.08, 95% CI 1.02–1.14), positive expectations for staying at work (OR 6.49, 95% CI 2.00–21.09), previous absenteeism for the same health complaint (OR 0.31, 95% CI 0.10–0.96) and good coping strategies (OR 1.13, 95% CI 1.04–1.23). For workers with other disorders, full work disability benefits, good mental health and positive expectations for staying at work were also prognostic factors for staying at work. Individual and policy factors seem to be important for staying at work of sick-listed workers with SHC and those with other disorders alike, but several biopsychosocial factors are particularly important for workers with SHC.

## 1. Introduction

Subjective health complaints (SHC) for which no pathological cause can be found after adequate physical examination are common in the general public and workforce of industrialized countries [1]. SHC is an umbrella term for health complaints (e.g., pain and dizziness) and syndromes (e.g., fibromyalgia and irritable bowel syndrome) that cannot be fully explained by a well-defined organic disease, comparable to other well-known terms such as medically unexplained physical symptoms and bodily distress disorder [2]. Approximately 30–70% of the working age population report at least one SHC during their working life [3,4]. In most cases, workers with SHC have only mild health complaints and can manage to stay productive at work, or they recover quickly and can return to work after a short period [3]. In 20–40% of the workers with SHC, however, the health complaints may become chronic, and the workers have persistent difficulties in meeting work demands [4,5]. This can lead to an increased risk of occupational dysfunction, long-term sickness absence and permanent exit from paid work [6].

Most research on workers with SHC has focused on identifying which workers are at increased risk of sickness absence, and on finding ways for absent workers to return to work [7,8,9]. This research has revealed that psychosocial and work-related factors in particular, such as mental distress, self-perceived disability, self-efficacy and expectations, social support, work demands, and compensation status, are associated with sickness absence and possibilities for returning to work [7,8,9]. Many researchers have argued that modification of these factors may help to prevent sickness absence and to support full return to work [10,11]. A key problem is that after long-term sickness absence, workers with SHC can partially return to work but may still experience difficulties in maintaining their work productivity and may be confronted with increased workload due to their chronic health complaints [12]. This group of workers therefore remains at increased risk for recurrent sickness absence and, ultimately, permanent exit from paid work [10,12]. As it is well known that early exit from paid work leads to a poorer quality of life [13], knowledge is needed on how to support staying at work for this group of workers.

To date, knowledge on factors that play a role in staying at work for workers with SHC is limited. The few studies that have examined work functioning and staying at work after return to work have mostly focused on well-defined chronic health complaints, or on a mixture of several chronic disorders [14,15], but not on SHC specifically. In many countries, it is difficult to examine long-term partially sick-listed workers with SHC, as the criteria for work disability benefits for this group of workers are mostly very strict. The conditions for work disability benefits in the Netherlands, however, do not distinguish between SHC and other disorders. We, therefore, investigated prognostic factors for staying at work for partially sick-listed workers with SHC who managed to stay at work (at least partially), as well as for workers with other disorders, to gain insight into which factors may be modified with timely interventions to avoid recurrent sickness absence after return to work and to determine whether these factors are different for workers with SHC and workers with other disorders.

## 2. Materials and Methods

### 2.1. Study Design and Study Population

We selected participants from the Forward cohort, which is a prospective cohort study performed among workers aged 18–65 years and registered as sick-listed for at least 84 weeks in the electronic database of the Dutch Social Security Institute between June 2014 and May 2015. The Forward cohort primarily aimed to find prognostic factors for return to work and included 2593 workers who met all inclusion criteria and returned a filled-in baseline questionnaire (T0) and a signed informed consent. We followed the included participants for 24 months with questionnaires after one year (T1) and two years (T2) from baseline. The flowchart in Figure 1 describes the design of the Forward cohort and the study population of the present study.

For the present study, we selected 519 participants from the Forward cohort who were still partially at work at baseline (*n* = 658), despite a medical condition (*n* = 595), and who had a fully documented work status during follow-up (*n* = 519). Information about work status was derived from the questionnaires, and information about the medical condition from the medical work disability assessments at the Dutch Social Security Institute, for which workers who are still sick-listed after 84 weeks can apply in the Netherlands. Insurance physicians, who perform these assessments, report diagnoses by using a code list [16], which is based on the International Classification of Diseases (ICD classification) [17]. If the insurance physician reported one of the 10 functional somatic syndromes (somatic (pain) syndrome; somatization disorder; pelvic girdle pain; tension headache; Tietze syndrome; irritable bowel syndrome; chronic fatigue syndrome; fibromyalgia; whiplash; and repetitive strain injury) or one of the 25 health complaints that matches with the 23 (partially) unexplained physical complaints of the Robbins list [18], then participants were indicated as having SHC (subjective health complaints). If the insurance physician reported another diagnosis, participants were indicated as having other disorders than SHC, and were used in the present study as a reference group.

### 2.2. Informed Consent

The Medical Ethics Committee of the Amsterdam University Medical Center (Vrije Universiteit Amsterdam; IRB00002991), gave ethical approval for the study. They declared that no comprehensive ethical review was needed for this study. All procedures performed in this study were in accordance with the ethical standards of this institutional research committee and with the 1964 Helsinki declaration and its later amendments or comparable ethical standards. All patients have given consent to the inclusion of material pertaining to themselves, and they were informed that we have fully anonymized all data so that their identity cannot be identified via the paper.

### 2.3. Measures

#### 2.3.1. Dependent Variable

The primary outcome measure was staying at work. Staying at work was assumed if participants, who were on long-term sickness absence, worked partially in a paid job at baseline (T0) and reported that they continued work participation in paid work, independent of the number of working hours, during the whole follow-up period (i.e., at T1 as well as T2). Participants who reported that they worked partially in a paid job at baseline, but not anymore during any of the follow-up measurements, were categorized as not staying at work.

#### 2.3.2. Independent Variables

The independent variables were collected from data of the Dutch Social Security Institute after the medical work disability assessment and via self-reported answers on general questions in the questionnaires at baseline (Appendix A) and validated questions in the questionnaires at baseline. We based the selection of variables on literature [7,8,14,15], and we used the biopsychosocial model to categorize the variables because it is a broad model that focuses on all aspects of functioning [17,19].

The validated questions were based on the following validated questionnaires:1.The work and well-being inventory (WBI) questionnaire with 85 questions and five subscales [20]:The stressors subscale with a scoring range of 16–64 (higher scores indicate more stressors).The support subscale with a scoring range of 21–84 (higher scores indicate better support).The symptom subscale with a scoring range of 20–80 (higher scores indicate more symptoms).The coping strategies subscale with a scoring range of 17–68 (higher scores indicate better coping).The self-perceived disability subscale with a scoring range of 7–28 (higher scores indicate more self-perceived disability).
2.The hospital anxiety and depression scale (HADS) with 14 questions and two subscales [21]:The depressive disorder subscale with a scoring range of 0–21 (higher scores indicate a higher risk for a depressive disorder).The anxiety disorder subscale with a scoring range of 0–21 (higher scores indicate a higher risk for an anxiety disorder).
3.The patient health questionnaire (PHQ-15) with 15 questions and one scale [22]:The severity of complaints scale with a scoring range of 5–30 (higher scores indicate more severe complaints).
4.The short form health survey 36 (SF-36) with 36 questions and three subscales [23,24]:The physical health subscale (PCS) with a scoring range of 0–100 (higher scores indicate better levels of physical health and functioning).The mental health subscale (MCS) with a scoring range of 0–100 (higher scores indicate better levels of mental health and functioning).The health change subscale (SF-2), which was derived from the following question on the SF-36: “How is your health in general compared to a year ago?” We categorized the five answering options into two categories: ‘same or better’ and ‘worse’.
5.The Whitely index questionnaire (WI) with 14 questions and one scale [25]:The hypochondria scale with a scoring range of 0–14 (higher scores indicate a higher risk for hypochondria).
6.The work ability index (WAI) with three questions and two subscales [26]:The work ability in general subscale with a scoring range of 0–10 (higher scores indicate higher self-perceived work ability).The work ability in the context of work load subscale with a scoring range of 2–10 (higher scores indicate higher self-perceived work ability in the context of work load).
7.The obstacles to return to work questionnaire (ORQ) with six questions and one scale [27]:The perceived prognosis of work return scale with a scoring range of 0–36 (higher scores indicate higher self-perceived possibilities for returning to work).

### 2.4. Statistics

We divided the participants into one subgroup with SHC and one subgroup with other disorders (reference group). We divided the independent variables into four domains (i.e., demographic, socio-economic and work-related, health-related, and self-perceived ability) based on the biopsychosocial framework [17,19]. For all variables, we analyzed the descriptives for both groups separately. We started further analyses with the SHC group. To analyze possible prognostic factors for staying at work for this group, we first checked for multicollinearity between the independent variables. Variables that had a variance inflation factor (VIF) of <10 and a Pearson correlation of <0.8 were included in the analyses [28]. For all included independent variables, we performed univariable logistic regression analyses, with the dependent variable staying at work. We performed multivariable logistic regression analyses with forward selection per domain separately with all independent variables that had a *p*-value ≤ 0.157 [29] in the univariable analyses. We used this Akaike information criterion of *p* ≤ 0.157 for the selection of predictors as it is widely used and also particularly recommended in the TRIPOD statement for a small data set [29,30]. Next, we combined all variables with a *p*-value ≤ 0.157 in the logistic regression analyses per domain in one multivariable logistic regression analysis with Forward selection. Subsequently, we analyzed all variables that remained with a *p*-value ≤ 0.05 in a combined final logistic model. To evaluate the overall fit and predictive ability, we analyzed the Hosmer–Lemeshow and Nagelkerke’s R2 Value of the final model [28]. We assessed the discrimination possibilities of the final model for the SHC group by applying the same final SHC model in the group with other disorders. We calculated the odds ratios (OR), 95% confidence intervals (95% CI), the Hosmer–Lemeshow and Nagelkerke’s R2 to compare the outcomes with the outcomes of the SHC group. We used SPSS version 24.0 and R-studio for all statistical analyses.

## 3. Results

Table 1 shows the baseline characteristics of the study population. A total of 86 workers with SHC (subjective health complaints) and 433 workers with other disorders (reference group) participated in the present study, with 44 participants (51%) in the SHC group and 242 participants (56%) in the reference group staying at work during the follow-up of two years. Overall, the baseline characteristics were comparable between the two groups, but in all four domains we found some differences between participants with SHC and those with other disorders. Participants with SHC were somewhat younger, more often female and less often the breadwinner of the family. They also had more psychologically than physically demanding jobs and received full work disability benefits less often than those with other disorders. Furthermore, participants with SHC tended to have more complaints and less self-perceived ability and positive expectations to function than participants with other disorders (Table 1).

### 3.1. Staying at Work Predictors for Participants with SHC

We included all independent variables in the univariable logistic regression analyses as we found VIF scores of <10 and correlations of < 0.8 for all variables and did not assume multicollinearity (Appendix A). Univariable logistic regression analyses showed 17 potential predictors (*p* ≤ 0.157) for staying at work, divided over all four domains (i.e., demographic, socio-economic and work-related, health-related and self-perceived ability) (Table 2). Multivariable logistic regression analyses with separate forward selection per domain showed that 11 of these 17 potential predictors remained statistically significant (*p* ≤ 0.157) (Table 3). We then combined these 11 potential predictors in one multivariable logistic regression analysis and found five statistically significant predictors (*p* ≤ 0.05) after forward selection, which we combined in the final model (Table 4). In this final model for workers with SHC, previous absenteeism for the same health complaint (OR 0.31, 95% CI 0.10–0.96) and full work disability benefits (OR 0.07, 95% CI 0.01–0.64) reduced the probability of staying at work. We also found that the chance of staying at work increased if participants reported a good mental health (OR 1.08, 95% CI 1.02–1.14), good coping strategies (OR 1.13, 95% CI 1.04–1.23) and positive expectations for staying at work (OR 6.49, 95% CI 2.00–21.09). We found a good fit for this final model: the Hosmer–Lemeshow was not statistically significant (*p*-value 0.57) and the Nagelkerke’s R2 was 0.51.

### 3.2. Staying at Work Predictors for Participants with Other Disorders (Reference group) than SHC

We applied the same variables of the final model for the SHC group to the group with other disorders and found statistically significant (*p* ≤ 0.05) associations with SAW for three out of the five variables (Table 4). In the socio-economic and work-related domain, we found that full work disability benefits (OR 0.13, 95% CI 0.08–0.21) reduced the probability of staying at work. Within the health domain, we found that if participants reported a good mental health (OR 1.03, 95% CI 1.01–1.05), they were more likely to stay at work. The domain of self-perceived ability showed that participants who reported positive expectations for staying at work (OR 3.15, 95% CI 2.00–4.97) stayed at work more often than those with negative expectations for staying at work. The Nagelkerke’s R2 was 0.33 and the Hosmer–Lemeshow was not statistically significant (*p*-value 0.66), indicating that there was also a good fit for the model for workers with other disorders than SHC.

## 4. Discussion

The primary aim of this prospective cohort study was to analyze prognostic factors for staying at work for partially sick-listed workers with SHC (subjective health complaints). The secondary aim was to analyze if these factors were also valid for partially sick-listed workers with other disorders. Our study showed that five factors across the biopsychosocial model were associated with staying at work for workers with SHC. We found that previous absenteeism for the same health complaint, poor coping strategies and full work disability benefits were negatively related to staying at work, and that a good mental health and positive expectations for staying at work were positively related to staying at work. Three of these five factors were also valid for workers with other disorders than SHC, which suggests that the mechanism underlying staying at work in workers with SHC are mostly comparable to those of workers with other disorders.

Although the present study was mainly based on workers with SHC who were able to work partially, eligibility for full work disability benefits still lead to a decreased chance of staying at work. The exact underlying mechanism that leads to this effect is difficult to extract directly from our results. As the severity of the complaints did not show a significant impact on staying at work, it seems unlikely that health status itself played a major role. Instead, an anti-therapeutic effect of full work disability benefits, as reported by Murgatroyd et al. [31], may play a role. Workers who receive full work disability benefits do not have the obligation to work and may fear losing their work disability status when staying at work. This concurs with the work of Cassidy et al. [32] and the OECD [33], which suggest that eligibility for full compensations is indeed associated with less work participation. Cassidy et al. [32] argue that this may be due to financial incentives or secondary gain, especially for workers with SHC as they may be more focused on proving that their health complaints are real. However, we found that a decreased chance for staying at work was also valid for workers with other disorders who were able to work partially but were also eligible for a full work disability benefit. This apparent contrast might be explained by an underlying mechanism: workers who are not eligible for compensation may effectively be forced to stay at work due to financial necessity, even if this exceeds their self-perceived work capacity and even if they have not recovered sufficiently [34]. Keeping in mind that we found that no or partial eligibility for work disability benefits positively impacted on staying at work, it should be possible to find a way in which partial work disability benefits can be granted that are better adapted to the individual needs and capacities of both SHC workers and workers with other disorders [33].

The importance to adapt to individual needs and capacities is further underpinned by our results, which indicate that a good mental health and positive expectations for staying at work are important factors for staying at work for both workers with SHC as for those with other disorders. This suggests that there is a possible relation between good mental health and positive expectations on the one hand and better capacities to deal with health complaints and meeting work demands on the other. This relation has also been addressed by other studies [7,8,35,36,37]. Some of these studies have even reported that the way in which workers respond and act in their rehabilitation process is largely based on good mental health and positive expectations, and they suggest that interventions focused on the individual capacities and needs in the working context decrease distress and may increase the mental capacity and expectations for workers at risk for sickness absence and permanent exit from paid work [35,36,37].

We found that workers with SHC with previous absenteeism for the same health complaint were less able to stay at work, which may suggest they are less able to deal with their complaints and to adjust to the specific demands of their job. We found that good coping strategies (e.g., good personal control) were associated with better possibilities for staying at work for workers with chronic SHC. It seems that those workers are better able to adjust to the specific demands of their job. A possible explanation is that those workers are better able to change cognitive and behavioral efforts and can adopt various strategies to deal with their complaints [38]. Our findings are consistent with those of a previous study that reported that non sick-listed women with fibromyalgia, who adopted successful strategies to cope with their problems, managed to continue to work without sickness absence [39]. Other studies have also showed that workers with effective coping strategies have better outcomes in their work functioning [8,40,41]. In addition, workers with good coping strategies seem to have a better self-efficacy, are more resilient and are better able to use past experiences to adapt their strategies [42]. Our results support the need for interventions aimed at enhancing coping skills (i.e. counselling programs and support systems) for workers with SHC, to improve their coping abilities and enhance their work ability and staying at work [43].

### 4.1. Strengths and Limitations

Our study design made it possible to evaluate the influence of work disability benefits over time on staying at work because we included workers after two years of sickness absence, just before their medical work disability assessment. The design of our study also made it possible to analyze the influence of these work disability benefits for workers with SHC and other common chronic disorders separately. The results from this Dutch cohort are useful for comparable Western countries whose legislation makes it especially difficult to examine workers with chronic SHC. Furthermore, our use of the biopsychosocial model [17,19] made it possible to study long-term effects of demographic, personal, health and work-related factors on staying at work equally, and gave us the opportunity to focus on all aspects and the synergy of multiple factors that play a role in work functioning and staying at work.

Unfortunately, our study only included a small number of workers, especially workers with chronic SHC. This could be a consequence of the manner in which we included participants and the fact that the Forward cohort primarily aimed to find prognostic factors for return to work. Via postal mail, we asked all registered sick-listed workers at the Dutch Social Security Institute whether they wanted to participate in the study, if they were still sick-listed and planning to apply for work disability benefits. We suspect that most workers who were partially sick-listed did not see themselves as sick-listed or were unsure if they would apply for a work disability assessment at all. Because we could only obtain information on work status and diagnosis after the work disability assessment, we could also not fully foresee the number of workers with SHC and other disorders. However, taking into account that approximately 15–20% of long-term sick-listed workers are sick-listed due to SHC [44], the distribution of workers in our study (SHC 17% and other disorders 83%) can be considered as representative. Still, the somewhat limited number of included workers is likely to have caused some selection bias. Unfortunately, we cannot obtain more information about the direction of bias, because data of workers who did not respond is unavailable due to privacy policies.

Additionally, there could be differences in the number of working hours between workers, also potentially leading to selection bias. Because of the use of self-reported outcome measures, it was difficult to compare hours at work. Therefore, we included all workers that were at work at baseline and at follow-up, irrespective of the number of hours at work. Despite the fact that questionnaires are valid and valuable sources of information, data gathered from objective registrations is preferable to data based on questionnaires [45]. However, we mostly used questionnaires to obtain information on predictor and outcome variables, and the sometimes incomplete questionnaires resulted in the exclusion of another 13% of the original participants. To assess if this biased our results, we performed a missing data analysis. Although workers with an unknown staying at work outcome differed in health compared to the workers with a known staying at work outcome, sensitivity analyses did not show any differences on regression coefficients in the final model. Therefore, we assumed that there is missing at random data and that the data in the complete case analyses is robust, unselective and representative for other workers [46]. We included only the results of the complete case analyses in this study; however, the missing data analyses, including the recommended missing data handling method multiple imputation, are presented in the Appendix A.

### 4.2. Implications for Policy, Practice and Future Research

To support partially long-term sick-listed workers with SHC for staying at work, our research suggests that stakeholders could focus on a multilevel solution. On the level of the individual worker, focus on the individual capacities and needs of the worker in the working context seems beneficial, with particular focus on improving self-management strategies and resilience of the worker. On a societal level, modifying the policies regarding the social security systems, particularly rules and regulations around work disability benefits, is advisable to avoid permanent exit from the workforce of workers with SHC. Further research is however needed to examine in which way these particularly rules and regulations have to be modified. It seems that this multilevel solution is also a good option for workers with other chronic disorders.

Other researchers have recommended comparable solutions for staying at work or returning to work, once workers are absent [33,43]. However, those recommendations and intervention studies are mostly based on a one-level solution and did not take into account the synergy of multiple factors [33,43]. More research is required to better examine the effect of a combination of supporting individual capacities and needs in the working context and modifying the policies of the social security systems for workers with several chronic disorders.

## 5. Conclusions

Staying at work for partially sick-listed workers with chronic subjective health complaints was associated with several biopsychosocial factors. We found similar factors for partially sick-listed workers with other chronic disorders. We therefore suggest a focus on multilevel solutions—supporting individual capacities and needs in the working context and modifying the policies of the social security systems—to support staying at work for sick-listed workers. Further research is needed to investigate in which way policy rules and regulations have to be modified and whether these suggested solutions can be implemented and evaluated in practice.

## Figures and Tables

**Figure 1 ijerph-17-07184-f001:**
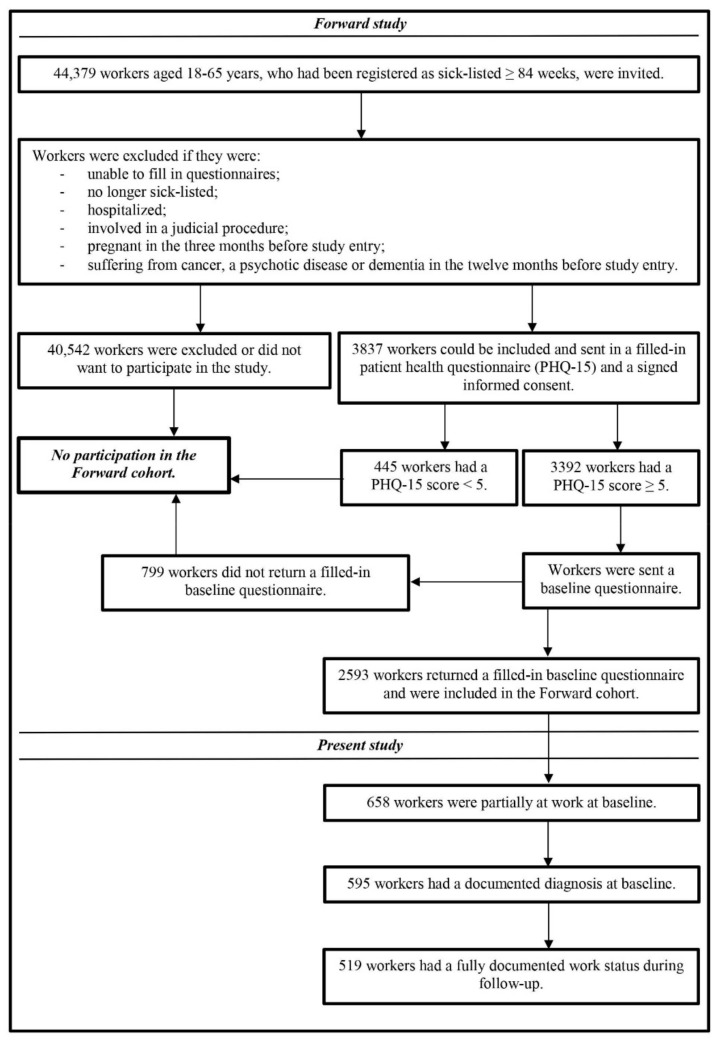
Flow chart of the study design of the Forward cohort and the study population of the present study.

**Table 1 ijerph-17-07184-t001:** Baseline characteristics of the study population.

		SHC ^1^ (No ^2^ = 86)		Other Disorders (No = 433)	
**Domains**	**Categories/Ranges**	**Mean/No**	**SD ^3^/%**	**Mean/No**	**SD/%**
**Demographic**					
Age in years	18–65	47.12	10.46	50.90	9.21
Gender	Male	29	34%	206	48%
Married or partner	Yes	69	80%	321	74%
Breadwinner of the family	Yes	49	57%	296	68%
Land of birth	The Netherlands	78	91%	393	91%
Educational level	Primary/Secondary school	33	38%	170	39%
	High school	27	32%	145	34%
	Bachelor’s/Master’s degree	26	30%	118	27%
**Socio-economic and work-related**					
Collar job	Blue	14	16%	105	24%
	White	31	36%	158	37%
	Pink	41	48%	170	39%
Employed	Yes	75	87%	378	87%
Usual working time in hours	4–60	30.84	8.31	33.20	8.69
Regular work schedule	Yes	59	69%	309	71%
Managerial position	Yes	15	17%	62	14%
Job demands	Psychological	20	23%	154	36%
	Physical	36	42%	132	30%
	Combination of both	30	35%	147	34%
Stressors^4^	16–64	35.84	9.11	35.64	8.39
Support^4^	21–84	60.79	10.49	61.41	11.38
Previous absenteeism for the same health complaint	Yes	39	45%	230	53%
Work disability benefits	No/Partial	73	85%	301	70%
Adjustments at work	Yes	70	81%	360	83%
Interventions at work (e.g., job coaching)	Yes	77	89%	392	91%
**Health-related**					
Use of specialist care in the last 2 years	Yes	73	85%	363	84%
Use of psychiatric care in the last 2 years	Yes	52	61%	213	49%
Use of medication	Yes	61	71%	368	85%
Depressive disorder^5^	0–21	7.60	4.10	7.66	4.62
Anxiety disorder^5^	0–21	7.40	4.06	8.07	4.17
Severity of complaints^6^	5–30	11.83	5.04	10.91	4.37
Physical health^7^	0–100	31.96	8.58	34.29	9.59
Mental health^7^	0–100	40.78	12.48	38.63	13.07
Health compared to a year ago^7^	Worse	28	33%	163	38%
Hypochondria^8^	0–14	5.28	2.93	5.38	2.98
Symptom scale^4^	20–80	41.01	9.20	41.37	10.39
Coping strategies^4^	17–68	40.68	9.37	41.17	9.51
**Self-perceived ability**					
Positive expectations for staying at work	Yes/Inconclusive	43	50%	271	63%
Disability^4^	7–28	21.70	4.65	20.54	4.90
Work ability in general^9^	0–10	4.14	2.04	4.64	1.96
Work ability in the context of work load^9^	2–10	5.82	1.52	6.15	1.50
Possibilities for returning to work^10^	0–36	15.00	8.85	14.77	9.54

Footnotes: ^1^ SHC = subjective health complaints; ^2^ No = number; ^3^ SD = standard deviation; ^4^ based on the work and well-being inventory questionnaire (WBI); ^5^ based on the the hospital anxiety and depression scale (HADS); ^6^ based on the patient health questionnaire (PHQ-15); ^7^ based on the short form health survey 36 (SF-36); ^8^ based on the Whitely index questionnaire (WI); ^9^ based on the work ability index (WAI); ^10^ based on the obstacles to return to work questionnaire (ORQ).

**Table 2 ijerph-17-07184-t002:** Univariable logistic regression analyses of all potential predictors for staying at work for participants with subjective health complaints (SHC).

Domains	Categories/Ranges	OR ^1^	95% CI ^2^	*p*
**Demographic**				
Age in years	18–65	0.99	0.95–1.03	0.51
Gender	Male	Reference	-	
	Female	0.97	0.40–2.37	0.94
Married or partner	No	Reference	-	
	Yes	0.92	0.32–2.65	0.87
Breadwinner of the family	No	Reference	-	
	Yes	1.19	0.51–2.81	0.69
Land of birth	The Netherlands	Reference	-	
	Other country	0.29	0.05–1.50	0.14
Educational level	Primary/Secondary school	Reference	-	
	High school	2.55	0.90–7.24	0.08
	Bachelor’s/Master’s degree	2.80	0.97–8.10	0.06
**Socio-economic and work-related**				
Collar job	Blue	Reference	-	
	White	1.62	0.45–5.78	0.46
	Pink	1.40	0.41–4.76	0.59
Employed	No	Reference	-	
	Yes	0.56	0.15–2.06	0.38
Usual working time in hours	4–60	1.03	0.98–1.08	0.30
Regular work schedule	No	Reference	-	
	Yes	0.77	0.31–1.93	0.58
Managerial position	No	Reference	-	
	Yes	1.11	0.36–3.39	0.85
Job demands	Psychological	Reference	-	
	Physical	1.02	0.34–3.07	0.97
	Combination of both	0.63	0.20–1.96	0.42
Stressors^3^	16–64	0.98	0.94–1.03	0.41
Support^3^	21–84	1.03	0.99–1.08	0.14
Previous absenteeism for the same health complaint	No	Reference	-	
	Yes	0.32	0.13–0.77	0.01
Work disability benefits	No/Partial	Reference	-	
	Full	0.06	0.01–0.47	0.01
Adjustments at work	No	Reference	-	
	Yes	1.44	0.48–4.30	0.51
Interventions at work (e.g., job coaching)	No	Reference	-	
	Yes	0.82	0.21–3.29	0.78
**Health-related**				
Use of specialist care for the last 2 years	No	Reference	-	
	Yes	0.88	0.27–2.88	0.83
Use of psychiatric care for the last 2 years	No	Reference	-	
	Yes	0.89	0.37–2.11	0.79
Use of medication	No	Reference	-	
	Yes	0.22	0.08–0.63	0.01
Depressive disorder^4^	0–21	0.89	0.80–0.99	0.05
Anxiety disorder^4^	0–21	0.94	0.85–1.05	0.28
Severity of complaints^5^	5–30	0.85	0.77–0.94	0.002
Physical health^6^	0–100	1.03	0.98–1.09	0.21
Mental health^6^	0–100	1.04	1.00–1.07	0.05
Health compared to a year ago^6^	Worse	Reference	-	
	Same/Better	3.21	1.24–8.32	0.02
Hypochondria^7^	0–14	0.82	0.70–0.97	0.02
Symptom scale^3^	20–80	0.96	0.91–1.00	0.07
Coping strategies^3^	17–68	1.05	1.00–1.10	0.06
**Self-perceived ability**				
Positive expectations for staying at work	No	Reference	-	
	Yes/Inconclusive	3.87	1.58–9.46	0.003
Disability^3^	7–28	0.92	0.83–1.01	0.08
Work ability in general^8^	0–10	1.25	1.00–1.56	0.05
Work ability in the context of work load^8^	2–10	1.36	1.00–1.85	0.05
Possibilities for returning to work^9^	0–36	1.02	0.97–1.07	0.54

Footnotes: ^1^ OR = odds ratio; ^2^ 95% CI = 95% confidence intervals; ^3^ based on the work and well-being inventory questionnaire (WBI); ^4^ based on the the hospital anxiety and depression scale (HADS); ^5^ based on the patient health questionnaire (PHQ-15); ^6^ based on the short form health survey 36 (SF-36); ^7^ based on the Whitely index questionnaire (WI); ^8^ based on the work ability index (WAI); ^9^ based on the obstacles to return to work questionnaire (ORQ).

**Table 3 ijerph-17-07184-t003:** Multivariable logistic regression analyses of 11 remaining potential predictors for staying at work for participants with subjective health complaints (SHC) per domain separately.

Domains	Categories/Ranges	OR ^1^	95% CI ^2^	*p*
***Demographic***				
Educational level	Primary/Secondary school	Reference	-	
	High school	2.55	0.90–7.24	0.08
	Bachelor’s/Master’s degree	2.80	0.97–8.10	0.06
***Socio-economic and work-related***				
Support^3^	21–84	1.04	0.99–1.09	0.11
Previous absenteeism for the same health complaint	No	Reference	-	
	Yes	0.33	0.13–0.87	0.03
Work disability benefits	No/Partial	Reference	-	
	Full	0.06	0.01–0.48	0.01
***Health-related***				
Use of medication	No	Reference	-	
	Yes	0.40	0.12–1.31	0.13
Severity of complaints^4^	5–30	0.90	0.79–1.03	0.12
Mental Health^5^	0–100	1.05	1.00–1.11	0.07
Health compared to a year ago^5^	Worse	Reference	-	
	Same/Better	2.77	0.87–8.80	0.08
Coping strategies^3^	17–68	1.08	1.01–1.15	0.02
***Self-perceived ability***				
Positive expectations for staying at work	No	Reference	-	
	Yes/Inconclusive	3.44	1.38–8.58	0.01
Work ability in the context of work load^6^	2–10	1.27	0.92–1.74	0.14

Footnotes: ^1^ OR = odds ratio; ^2^ 95% CI = 95% confidence intervals; ^3^ based on the work and well-being inventory questionnaire (WBI); ^4^ based on the patient health questionnaire (PHQ-15); ^5^ based on the short form health survey 36 (SF-36); ^6^ based on the work ability index (WAI).

**Table 4 ijerph-17-07184-t004:** Final model of all remaining predictors for staying at work for participants with subjective health complaints (SHC) and other disorders separately.

		SHC (No ^1^ = 86)			Other Disorders (No = 433)		
**Domains**	**Categories/Ranges**	**OR ^2^**	**95% CI ^3^**	***p***	**OR**	**95% CI**	***p***
**Socio-economic and work-related**							
Previous absenteeism for the same health complaint	No	Reference			Reference		
	Yes	0.31	0.10–0.96	0.04	0.72	0.46–1.13	0.16
Work disability benefits	No/Partial	Reference			Reference		
	Full	0.07	0.01–0.64	0.02	0.13	0.08–0.21	0.000
**Health-related**							
Mental Health^4^	0–100	1.08	1.02–1.14	0.01	1.03	1.01–1.05	0.002
Coping strategies^5^	17–68	1.13	1.04–1.23	0.004	1.02	1.00–1.04	0.23
**Self-perceived ability**							
Positive expectations for staying at work	No	Reference			Reference		
	Yes/Inconclusive	6.49	2.00–21.09	0.002	3.15	2.00–4.97	0.000

Footnotes: ^1^ no = number; ^2^ OR = odds ratio; ^3^ 95% CI = 95% confidence intervals; ^4^ based on the short form health survey 36 (SF-36); ^5^ based on the work and well-being inventory questionnaire (WBI).

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
