# Peer review of "Prognostic Factors for Staying at Work for Partially Sick-Listed Workers with Subjective Health Complaints: A Prospective Cohort Study."

_ijerph, 2020, doi:10.3390/ijerph17197184_

Round 1

Reviewer 1 Report

The study examined predictors of staying at work among partially sick listed persons suffering from subjective health complaints, referring to various health conditions for which a specific pathological cause cannot be found, and compared these to the predictors of those with specified health problems. Among 157 possible predictors, having full work disability benefits, good mental health, having positive expectations for staying at work, having been previous absent due to the same health complaint, and having good coping strategies predicted staying at work over two years. Full work disability benefits, good mental health and positive expectations for staying at work predicted also staying at work among participants with specified health problems.

Overall, I find the study very sound and clear. The study was based on quite low number of participants (especially those with subjective health complaints) and these seems to be little information on their work conditions but the study has also several strengths, including a very large number of measurements based on validated scales, profound knowledge on the subject, and clear reporting style.

I have only very small comments on the manuscript.

I was a bit disturbed by the very abundant use of abbreviations. I’m not sure if some of these abbreviations such as SAW, are well-established or commonly used. I would suggest discarding some of the abbreviations

Another wording that bothered me was the expression “did not meet the exclusion criteria” which seems like a double negation.

Among the initial 172 predictors 17 was first selected based on univariate analysed and then these were further restricted to 11 after a domain specific analyses. To me this domain-specific step seems unnecessary as the variables in each specific domain are probably not very much correlated. However, I do not insist omitting it.

Predictors with a p-value ≤0.157 were selected for further analyses. Please use a couple of word for selecting this boundary.

Author Response

Point 1: I was a bit disturbed by the very abundant use of abbreviations. I’m not sure if some of these abbreviations such as SAW, are well-established or commonly used. I would suggest discarding some of the abbreviations.

Response 1: We thank the reviewer for this comment and agree that some of the abbreviations are probably not well-established and therefore possibly not clear to all readers. We have now discarded the following abbreviations; SAW, MUPS, BDD, RTW, UWV, IC, CAS and replaced them for full term text in the Revised Manuscript.

We did not change the abbreviation for the questionnaires, as we adopted these abbreviations from the developers of the questionnaires and we considered them as well-known by all readers.

And we also did not change the abbreviation for subjective health complaints (SHC) as this is a commonly used term in occupational health literature (Ihlebaek C, Eriksen HR. Occupational and social variation in subjective health complaints. Occup Med (Lond). 2003;53(4):270-278. doi:10.1093/occmed/kqg060), psychological literature (Versluis A, Verkuil B, Brosschot JF. Reducing worry and subjective health complaints: A randomized trial of an internet-delivered worry postponement intervention. Br J Health Psychol. 2016;21(2):318-335. doi:10.1111/bjhp.12170) and public health literature (Fridh M, Lindström M, Rosvall M. Subjective health complaints in adolescent victims of cyber harassment: moderation through support from parents/friends - a Swedish population-based study. BMC Public Health. 2015;15:949. Published 2015 Sep 23. doi:10.1186/s12889-015-2239-7).

Point 2: Another wording that bothered me was the expression “did not meet the exclusion criteria” which seems like a double negation.

Response 2: We agree with the reviewer that the expression of this phrasing is not optimal and might be confusing for readers. We have now adapted the text in the Revised Manuscript in the Materials and Methods section under the subheading ‘Study design and study population’ (lines 82-85, page 2):

“The Forward cohort primarily aimed to find prognostic factors for return to work and included 2,593 workers who met all inclusion criteria and returned a filled-in baseline questionnaire (T0) and a signed informed consent.”

We have also adapted Figure 1 (lines 104-106, page 3). We hope that this makes it more clear.

Point 3: Among the initial 172 predictors 17 was first selected based on univariate analysed and then these were further restricted to 11 after a domain specific analyses. To me this domain-specific step seems unnecessary as the variables in each specific domain are probably not very much correlated. However, I do not insist omitting it.

Response 3: We thank the reviewer for pointing this out. Although we certainly understand the comment of the reviewer, we have chosen not to omit this step in the analyses because we want to be as clear as possible to the readers of the steps taken in our analyses. Furthermore, we prefer to follow the statistic rules as much as possible (Steyerberg, E.W. Clinical Prediction Models: A Practical Approach to Development, Validation, and Updating. 1st ed. New York: Springer; 2009), (Field, A. Discovering statistics using IBM SPSS statistics. 4th ed. London: Sage Publications Ltd; 2015) and (Moons, K.G.; Altman, D.G.; Reitsma, J.B.; Ioannidis, J.P.; Macaskill, P.; Steyerberg, E.W.; et al. Transparent reporting of a multivariable prediction model for individual prognosis or diagnosis (TRIPOD): explanation and elaboration. Annals of internal medicine 2015, 162(1), W1-73.).

Point 4: Predictors with a p-value ≤0.157 were selected for further analyses. Please use a couple of word for selecting this boundary.

Response 4: We thank the reviewer for this comment and agree that the selection of this boundary was not clearly described in the manuscript. To clarify this more, we have added the text of the Revised Manuscript in the Materials and Methods section under the subheading ‘Statistics’ (lines 178-180, page 5):

“We used this Akaike information criterion of P≤0.157 for the selection of predictors as it is widely used and also particularly recommended in the TRIPOD statement for a small data set.”

We have also included a new reference in the reference list to support the above addition (lines 494-495, page 14):

“30. Steyerberg, E.W. Clinical Prediction Models: A Practical Approach to Development, Validation, and Updating. 1st ed. New York: Springer; 2009.”

Reviewer 2 Report

This is an observational study using the Forward cohort to determine the prognostic factors for staying at work for partially sick-listed workers with subjective health complaints. The authors identified 5 biopsychosocial prognostic factors using the forward stepwise regression models, including previous absenteeism for the same health complaint, availability of full work disability benefits, mental health status, coping strategies, and expectations for staying at work.

Major comments:

  1. Abstract: There is a lack of information about the study cohort. Provide additional information about the study source and some baseline characteristics (for examples age, gender, health status).
  2. Methods, 2.2.2 Independent variables: Information on general questions and collected data (Line 115 to 126) would be better presented in a supplemental table format
  3. Provides multicollinearity analyses as a supplemental table.
  4. For tables 2 to 4, the questionnaires used to assess each independent variables listed are unclear. Need to add footnotes to specify the questionnaires used for the independent variables, where applicable.
  5. For tables 2 to 4, add p-values instead of bolding to allow easier interpretation.

Author Response

Point 1: Abstract; There is a lack of information about the study cohort. Provide additional information about the study source and some baseline characteristics (for examples age, gender, health status).

Response 1: We thank the reviewer for this comment and agree that a more detailed description of the study and source population is needed. We have now added information about the study source and the baseline characteristics of the study population in the Revised Manuscript in the Abstract section (lines 19-23, page 1):

“We used data of 86 partially sick-listed workers with SHC (57 females, 29 males, mean age 47.1 years) and 433 with other disorders (227 females, 206 males, mean age 50.9 years), from an existing prospective cohort study consisting of 2,593 workers aged 18-65 years and registered as sick-listed with different health complaints or disorders for at least 84 weeks in the database of the Dutch Social Security Institute.”

Point 2: Methods, 2.2.2 Independent variables: Information on general questions and collected data (Line 115 to 126) would be better presented in a supplemental table format.

Response 2: We thank the reviewer for pointing this out. We have now transferred the text on the general questions and collected data from the Dutch Social Security Institute to a supplementary table and we have incorporated a reference to this supplementary file in the Revised Manuscript in the Materials and Methods section under the subheading ‘Independent Variables’ (lines 124-126, page 4).

“The independent variables were collected from data of the Dutch Social Security Institute after the medical work disability assessment and via self-reported answers on general questions in the questionnaires at baseline (Table S1) and validated questions in the questionnaires at baseline.”

Point 3: Provides multicollinearity analyses as a supplemental table.

Response 3: We think indeed that it is important to give more insight into the multicollinearity analyses by providing supplemental material. For the analyses of the correlation scores we have added a figure instead of a table. We think this gives more clarity than a table because of the amount of variables. For the outcomes of the VIF analyses we have added a supplemental table. We have incorporated references to both supplementary files in the Revised Manuscript in the Results section under the subheading ‘Staying at work predictors for participants with SHC’ (lines 215-217, page 7).

“We included all independent variables in the univariable logistic regression analyses as we found VIF scores of <10 and correlations of <0.8 for all variables and did not assume multicollinearity (Figure S1 and Table S2).”

Point 4: For tables 2 to 4, the questionnaires used to assess each independent variables listed are unclear. Need to add footnotes to specify the questionnaires used for the independent variables, where applicable.

Response 4: We agree with the reviewer that the variables listed in the tables are not completely clear. We have now added footnotes to specify the questionnaires used for the independent variables underneath all tables where applicable, including the supplementary tables.

Point 5: For tables 2 to 4, add p-values instead of bolding to allow easier interpretation.

Response 5: We thank the reviewer for pointing this out. We have adjusted the tables in line with the reviewer’s comment in all tables if applicable in the Revised Manuscript, including the supplementary tables.

Round 2

Reviewer 2 Report

The authors have addressed all comments appropriately.